# Analysis of 5G and LTE Signals for Opportunistic Navigation and Time Holdover

**DOI:** 10.3390/s24010213

**Published:** 2023-12-29

**Authors:** Adrian Winter, Aiden Morrison, Nadezda Sokolova

**Affiliations:** 1Department of Engineering Cybernetics, Norwegian University of Science and Technology (NTNU), O. S. Bragstads Plass 2D, 7034 Trondheim, Norway; adrian.winter@ntnu.no; 2SINTEF Digital, Strindvegen 4, 7034 Trondheim, Norway; nadia.sokolova@sintef.no

**Keywords:** navigation, timing, 5G, LTE, cellular, signals of opportunity

## Abstract

The purpose of this study was to evaluate the stability and therefore suitability of available fifth generation (5G) and long-term evolution (LTE) signals for positioning navigation and timing (PNT) purposes with particular focus on answering questions around the time-scale-dependent stability of these sources, which, to our knowledge, has not been addressed in the context of the numerous publications within the PNT community to date. The methodology used directly measured the over-the-air signal phase stability to one or more of the cellular signal sources that were visible from the lab environment simultaneously while using a local atomic clock or differential measurements to isolate the time stability of the observable cellular downlink signals. This approach was taken since it does not require subscription or association with the networks under test. Instead, it exploits a ‘signal of opportunity’ (SoP) approach to signal use for PNT purposes. The somewhat surprising result is that the time domain instability of the sources was highly variable, dependent on the implementation choices of the operator, and that the stability of even the modernized towers was generally best at interrogation intervals of approximately 0.01 s, which indicates that the existing exploitation of these signals within the PNT community has substantial room for improvement through simple changes to the selected update rate used.

## 1. Introduction

While global navigation satellite systems (GNSS) are the preferred options for many positioning navigation and timing (PNT) users and applications, the signals that reach terrestrial users have power levels of approximately −130 dBm. Therefore, they are ill suited for penetrating deep into buildings and are susceptible to even unintentionally generated radio frequency interference (RFI) [1]. To provide alternative and fallback navigation capability to users who must continue to navigate or synchronize their systems when GNSS signals are unavailable, a large body of research has been dedicated in recent years to the exploitation of Signals of Opportunity (SoP) including, but not limited to, television, cellular [2,3,4,5,6,7,8], low earth orbit (LEO) satellite [9], and other emitters of opportunity [10] that can be used as ersatz PNT observables. Cellular signals in particular stand out as excellent candidates for three very important reasons. First, the signals have much higher received power levels due to the much shorter ranges between the transmitters and devices within their coverage area (even penetrating deep into buildings by design unlike most signals transmitted from navigation or communication satellites). Second, the infrastructure in urban areas and along roadways, where GNSS RFI is relatively common, is dense by virtue of the market pressures for multiple network operators to cover entire cities and transport routes. Third, the third-generation partnership project (3GPP) standard organizations have proposed features within newer cellular standards, such as fifth generation (5G), which are directly intended to support navigation. While 5G navigation support might sound ideal, the primary operating mode foreseen is for the network operator to collect information from the user and to return their position information as a service, such as in past network time difference of arrival (TDOA) methods [11]. This makes it unfeasible to combine observations from competing network operators, as well as makes it unclear whether the various operators will deploy dense enough networks to enable practical navigation when using only their infrastructure. As an additional complication, the use of the term 5G has become more of a marketing term than a technical one in that user devices may report 5G coverage while the underlying signal structure remains almost indistinguishable from the previous generation of long-term evolution (LTE) coverage in that area. 

To investigate the suitability of the locally available LTE and 5G signals for SoP and synchronization purposes, the authors undertook observation and analysis of the availability, stability, and compatibility of the signals transmitted by local network operators in the vicinity of the facilities of SINTEF and the Norwegian University of Science and Technology (NTNU) in Trondheim, Norway. The results of this study indicate that, while the supposed 5G signals often implement only a small subset of the possible enhancements discussed in the context of 5G evolution, the signals transmitted by the base stations have characteristics that are very promising for use as both navigation and timing constraints. The novel contribution of this study to the existing body of literature involves the time scale-dependent analysis of the stability of the signal sources in both single-network-sourced and multi-operator modes, which helps inform what measurement and averaging intervals may be most appropriate for future PNT-focused users of 5G signals.

While knowledge about cellular base station clock stability is of limited value taken by itself, especially since no explicit time reference is embedded in the signal structure, the core novelty of this work is in demonstrating that this information is in fact very valuable for exploiting these signals in the domains of PNT. When investigating opportunistic navigation and time keeping for practical purposes, one should not assume the best case (no remote clock error) nor the worst case (maximum permissible clock error according to the specification) but, where possible, employ an interval specific stability metric to inform both the measurement update rate employed and the associated instability parameters expected. 

## 2. Materials and Methods

The system used for data collection comprised one or two Ettus Research (Ettus Research, a National Instruments Brand. 11500 North Mopac Expressway. Austin, TX 78759, USA) Universal Software Radio Peripheral (USRP) devices (model B200/210), which were connected to a multiband cellular antenna and linked to a computer via USB3 for data streaming, processing, and storage. The measurements and observables were generated by the real-time *SofTwAre REceiver* (STARE), which is a software-defined radio (SDR) that was specifically developed for measurement and observable generation from LTE and 5G new radio (NR) cellular downlink signals [12,13]. A diagram of the test setup can be found in Figure 1. The specific software version of STARE used in this study was v.1.4.0, which was, in turn, linked with a UHD, version 4.1.0.5-3.

The observables generated by STARE were subsequently processed with a custom script written in Julia, which visualizes the observables, performs statistical analysis, and calculates the Allan Deviations, as described in Section 3.4, using the software library AllanDeviations.jl v.0.3.0 [14]. 

To maintain a stable local time reference and minimize the measurement errors introduced by local clock instability, a GNSS-disciplined oscillator was used in each test. While, by default, the USRP models used can integrate an ovenized quartz oscillator (OCXO) with GNSS disciplining, we, in this study, elected to use an external rubidium (Rb) atomic clock with GNSS disciplining. In the former case, the OCXO oscillator was an Ettus research GNSS-disciplined OCXO, while, in the latter case, the time reference was the Safran (Safran Electronics & Defence, 72–76 Rue Henry Farman, 75015 Paris, France) LNRClok-1500 GNSS-disciplined Rb atomic clock. In all of the tests discussed in this paper, measurements were taken from static base stations while the reception antenna was also kept static throughout the data collection process.

All data presented in this study were recorded from the author’s offices in Strindvegen 4 using only live signals. No simulated data were used. 

### 2.1. Tracked Signals

STARE operates on the primary and secondary synchronization signals (PSS and SSS, respectively) embedded in LTE and 5G signals. These synchronization signals are broadcast by towers as a fundamental part of the signal structure and can be used without any form of authentication or subscription, thus making them particularly suitable for SoP purposes. The signals are, for the most part, known a priori, and they significantly help in the detection of available towers, as well as in initial signal acquisition. The fact that we operated in a fully non-authenticated state also implied that any operator- or situation-specific effects (such as the number of connected devices, current network utilization, operator specific settings, type and quantity of data traffic, quality of service settings, etc.) had no impact on the measurements. 

The only factor the authors believed to have a potential effect on the measurements were the base station power saving measures. Since the base stations cannot suspend the transmission of PSS and SSS without becoming invisible to the user equipment (UE), they generally do not do so outside of specific situations, as discussed in Section 4.1. 

All cellular transmissions in Norway work in frequency division duplex (FDD) mode, and—while the bandwidth of signals varies—the PSS and SSS features are, in all cases, constrained to a subset of the central 63 (LTE [15]) or 240 (5G [16]) orthogonal frequency division multiplexing (OFDM) subcarriers. 

### 2.2. The Test Setup and Signal Propagation Environment

All results presented here were recorded from the authors’ offices in Strindvegen 4, Trondheim, Norway, from which multiple base stations were available for measurements, as documented in Table 1 Within this study, we present the data from a subset of these visible base stations, which could be reliably tracked over the extended periods required for extended stability analysis without loss of lock. 

Unfortunately, due to national security concerns [17], the locations of these base stations are no longer public information, and the authors are therefore unable to conclusively say which signal was transmitted from which position. Consequently we also cannot say if there are any shared characteristics such as base station equipment or antennas between any of the cells. We visually identified at least five base station clusters less than 300 m from the reception antenna location and believe that all of the signals discussed here emit from one of those base stations. 

The RF propagation environment is urban and contains several large concrete buildings that are expected to produce a complex but mostly static propagation environment. Two of the five mentioned base stations are directly visible from the authors’ office, while three of them are obstructed by a building. Only moving vehicles on the adjacent roadway were suspected as significant non-static reflectors. 

### 2.3. Parameter Set Used for Recordings

The common settings found in Table 2 were used for recording all data sets. The cell-specific settings, which are the carrier frequency and the physical cell ID, can be found in Table 1. All of the parameters that are not specifically mentioned here are the default parameters used by STARE and can be found in the manual [18]. The sample rate is the product of FFT size and subcarrier spacing; thus, it is indirectly specified by the information contained in Table 2. 

**Table 1 sensors-24-00213-t001:** Adjacent to the NTNU campus infrastructure from all three network operators, a variety of signals spanning approximately 2 GHz of a spectrum are visible.

Operator ^1^	DL Frequency (MHz)	Band Number	Bandwidth (MHz)	Cell PhyID ^2^
Telenor	816.0	20	10	94
Telenor	955.0	8	10	411
Telenor	1815.6	3	10	466
Telenor	1830.0	3	20	214
Telenor	2140.0	1	20	275
Telenor	2660.0	7	20	214
Telenor	2679.8	7	20	429
Telia	935.1	8	10	3
Telia	935.1	8	10	169
Telia	1850.0	3	20	169
Telia	2120.0	1	20	385
Telia	2630.0	7	20	272
ICE	465.0	31	5	101
ICE	796.0	20	10	19
ICE	1870.0	3	20	19

^1^ Based on the publicly available spectrum allocation at the time of publication [19]. ^2^ Physical ID.

## 3. Results

### 3.1. Site Survey

The first step of the data collection process was a survey of the available cellular base stations visible from the chosen antenna location, the results of which are placed in Table 1, where the ID of each visible cell is presented along with the associated downlink carrier frequency and operator information. 

### 3.2. Visible Cell Stability

While it is known that cellular infrastructure must meet certain minimum time and frequency stability specifications, which have generally become more stringent as data rates and network densities have increased, the implementation details can leave room for interpretation on matters such as which time scale a network may be synchronized to [20,21]. Based on a survey of the existing literature, the stability of the time source within the base stations could be as poor as a normal quartz oscillator [22], but it is likely steered to a common time scale that is shared among the adjacent stations that are operated by the same network operator to prevent intercell interference and to meet the synchronization requirements of 5G [23]. To test this assumption directly, the data from the observed stations were gathered over periods of multiple hours using the STARE SDR. Internally, the STARE SDR tracks the epoch of arrival of signal features, including the primary and secondary synchronization sequence (PSS and SSS, respectively), versus the sampling epochs of the radio receiver, as well as converts these measurements to range deltas by multiplying the time observation with the speed of light. By smoothing these with the signal-carrier phase, a low noise estimate of the change in apparent range between the base station antenna and the user antenna is produced. Since both the transmit and receive antennas were static in these tests, the remaining measurements were, due to noise, clock deviations at the source and receiver; in addition, they also produced multipath effects. While multipath effects for static users of GNSS are not constant over long time periods due to the changing relative user–reflector–satellite geometry [24], the multipath between a static user and a static transmitter should be mostly constant (ref. Section 2.2 for further discussion), thus leaving primarily noise and clock changes in the resulting delta range measurements. 

The measured range deltas for Cell ID 214, 2660.0 MHz (Telenor) and Cell ID 19, 1870.0 MHz (ICE) are presented below in Figure 2a,b, respectively.

While the results shown in Figure 2 seem to indicate that the two stations observed implement timing solutions with very different levels of stability, it is not yet possible to determine if this is true generally for a given network operator and service type or if this is simply anecdotal to this realization of the transmitter. During a very long segment of about ten hours of tracking the former tower, it appeared as if the timing solution was very stable even over longer periods (see Appendix B). However, to gain confidence in the relative synchronization results, it was decided to run additional tests using multiple USRP radios to monitor pairs of cells that were operated by two network service providers simultaneously to test whether the instabilities evidenced above were representative and common in mode to all local cells on the same network or whether they differed between them.

The measurements presented in Figure 2b show a discontinuous rate with clear corners in the slope of the curve of phase measurement, which indicates that the clock in the cell was changing or being steered aggressively with long pauses between updates. This behavior represents a clear challenge to the exploitation of the signal for PNT purposes as time-varying error characteristics can frustrate online estimation. However, even if the other cells operated by the same network service provider are as unstable, they could still be well disciplined to a common clock, thus allowing for relatively low-noise TDOA or pseudo time of arrival (TOA) observable production and navigation techniques, such as those discussed in [25,26,27,28,29], to be applied with acceptable performance. 

### 3.3. Differential Stability Measurement

Due to the constraints of the radios available and the SDR software in use, differential measurements between multiple cells were generally only possible when two or more radios were employed to allow for the simultaneous measurement of multiple downlink channels. Since this involved running multiple instances of the SDR and relying on the accuracy of the timestamps of the generated observables as inputs, it was considered essential to test the stability of this process as independent of the other variables.

#### 3.3.1. Evaluation of Multi-Radio Multi-SDR Observable Stability

To isolate all the other parameters, two radios were set to observe the same cell simultaneously while connected to the same antenna and driven by the same local Rb oscillator, a configuration that is analogous to a ‘zero-baseline’ differential measurement in GNSS systems [30]. This configuration would ideally always produce a null output value, but some level of differential thermal noise and other parameters will also be present in real systems. Over multiple one-hour periods of observation, the peak-to-peak variation in the differential measurement was less than 0.02 m. Since this result was more than an order of magnitude smaller than the observations obtained from either individual cellular downlinks that were measured previously, it was considered safe to neglect this factor in subsequent analyses. The results of these tests are placed in Appendix A.

#### 3.3.2. Evaluation of the Differential Tower-to-Tower Stability

The main motivation for extending the analysis to include tower-to-tower differential measurements was to further isolate and eliminate any potential contributions of the local clock used to drive the SDRs, which, despite being a GNSS-disciplined atomic clock, will also have instabilities. While the results in Figure 2a show that the differences between the transmitter and receiver time scales were limited to less than 5 ns over this hour of observation, the specification provided by the manufacturer of the GNSS-disciplined Rb clock only stated a performance guarantee of 50 ns, thus causing this uncertainty to dominate the results. To remove this uncertainty, we resorted to observing two cells simultaneously via radios driven by a common oscillator, two pairs of which are plotted in Figure 3.

To eliminate the contribution of our local oscillator error, we needed only to differentiate the measurement series, which resulted in the traces shown in Figure 4, where the behavior of the differential cell site clocks was isolated.

Once the impact of the local clock instability was eliminated through the differential measurement process, the relative phase shift between the two measured cells exposed clear behavioral differences between the first and second combinations. The first combination of the two sources that were operated by the same service provider showed a nearly three times lower peak-to-peak relative variation in the TDOA than that which was formed between the two cells that were operated by different service providers. This is easily explicable as being a consequence of the expectation that two proximate cells operated by the same service provider are likely to be tightly steered to a common reference time scale when compared to infrastructure that is operated by different service providers. In the former case, the transmitters are potentially steered to a common physical timing device, while in the latter case the networks may or may not even be steered to the same realization of a common time scale, such as in the case with equipment that steers to the GPS realization of UTC (USNO) versus the European Galileo realization UTC (GST).

The visible step changes in the delta between the two towers operated by the same network operator could indicate a phase multipath reflection effect, or they could be real shifts in the phase of one or both transmitters. Meanwhile, by contrast, the process between the two dissimilar service providers appeared dominated by ramp behaviors with a small range of slew rates. On cursory investigation, the nearly three times better (smaller) peak-to-peak delta variation shown in Figure 4a compared to that shown in Figure 4b suggests it is the better combination for use as a PNT measurement pair. However, a deeper analysis showed that this was not actually the case for most measurement rates, as will now be shown.

### 3.4. Scale-Dependent Stability

While it is typical to express uncertainty levels in a navigation observable using a standard deviation at an arbitrary observation interval, within the domain of timing and time stability, it is nearly universal that the instability should be expressed at a specific measurement interval or period. The reason for this is that clocks are systems from which measurements may be used or could be useful over several orders of magnitude of measurement intervals, from precision interval measurement at fractions of a second to extended time hold overs for periods of days or longer. Depending on which noise processes are present in the device under test, their relative intensity, the observation interval, and the level of timing uncertainty can also span orders of magnitude—even when expressed as a fraction of the interval rather than as an absolute time error. Typically, over ‘short’ time scales, a given timing source will be dominated by white measurement noise, meaning that measuring for longer or averaging a longer interval results in a monotonically decreasing uncertainty with a slope near −1:1 when plotted on a log–log graph of instability versus time. Within some system-dependent averaging intervals, however, one or more other noise processes will come to dominate the result, typically starting with flicker noise from system electronics, or in random walk phases or frequency noises, which make further averaging counterproductive. 

One of the traditional ways of expressing this time-scale-dependent uncertainty is the Allan variance [31], which is calculated using Equation (1):(1)$$\sigma_{y}^{2}(\tau )=\sum_{i=1}^{M-1}{\left[x_{i+1}-x_{i}\right]}^{2},$$

The non-overlapped Allan variance or deviation has been generally superseded by the overlapped modified Allan deviation or the Hadamard deviation due to their better use of data, as well as due to their insensitivity to common time-keeping device errors, which are trivially compensated such as in constant linear drifts. Here, we have elected to use the modified Allan deviation [14] as it is better suited to exposing the transition between flicker phase noise and other processes. It was implemented via Equation (2) and applied to the phase measurements in units of meters.
(2)$$Mod{\;\sigma }_{y}^{2}(\tau )=\frac{1}{2m^{2}\tau^{2}\left(N-3m+1\right)}\sum_{j=1}^{N-3m+1}{\left(\sum_{t=j}^{j+m-1}\left[x_{t+2m}-{2x}_{t+m}+x_{t}\right]\right)}^{2}$$

When the differential cell measurement time series, shown in Figure 4a,b, were converted to the modified Allan deviations plotted in Figure 5, the time domain variation was converted into a time-scale-dependent uncertainty parameter, where different noise processes manifested at different time scales and in different proportions. A counterintuitive result immediately emerged showing that the pair that produced the far worse (larger) peak-to-peak variation was, in fact, more stable at most time scales than the less variable station pair.

By inspecting these curves, we can extract very useful information about the relative stabilities of the measurements produced, as well as how these vary with increasing observation intervals or measurement averaging. In Figure 5, the series that had the worse peak-to-peak variation (Figure 4b and Figure 5b) was statistically more stable over all measurement time scales that were less than 9 s of interrogation, except for a small region around a 1 Hz measurement. Meanwhile, both stations showed a ‘knee’ in their noise performances around 10^−2^ s or 100 Hz update rates. The high-frequency position of this knee indicated that anyone using these TDOA measurements, whether for PNT purposes or otherwise, would achieve optimal measurement performance by using the data at these high rates rather than coalescing or averaging over longer intervals such as 1 s, which may be adopted by default in some implementations. The authors believe that this observation represents a novel observation for this form of measurement in the context of PNT and SoP exploitation of 5G and other cellular signals. Additional results are provided in Appendix B. 

## 4. Discussion

When formulating this study, one of the open questions was to what extent the locally visible infrastructure had implemented 5G in accordance with the available specifications. Indeed, the user phones announce 5G coverage, yet when a radio reporting app is used to extract the parameters of the base station to which the device has associated, an LTE link status is given. Over the previous years, there have been lawsuits in the United States where service providers have been sued for deceptive advertising of the extent or capability of their 5G coverage. But regardless of the outcomes, the authors believe that part of the confusion arises from the large amount of flexibility in the definitions of what a 5G signal is or can be [32]. The publicly available specifications indicate that 5G is a multidimensional set of optional features and configurations, which might explain the inconsistent link status reporting, as well as the observed variability in the source timing stabilities and solutions implemented by the network service providers as measured in this study.

### 4.1. Downlink Signal Availability and Power Levels

One of the unexpected observations in this study was the large dynamic range of the reported received signal-to-noise ratio (SNR) reported by the STARE SDR over extended observation intervals, which did not seem to be accompanied by a change in the noise statistics of the measured delta range values. This observation implies that the downlink signals from the towers are dynamically adjusted during periods of light load, but not in a manner that the authors can conclusively identify from either the literature or observations. The literature supports a plurality of power saving options for 5G in general, though the vast majority of these are focused entirely on the power savings within UE devices [33,34], which is reasonable as these tend to be battery powered. Even in cases where discussions of modification to the downlink signal structure are explicitly discussed in the context of energy savings, such as in [35], adjusting the base station multiple input multiple output (MIMO) configuration dynamically is typically for optimizations intended to reduce UE energy requirements. The 3GPP standards do explicitly allow for a dynamic capacity scaling of a 5G gNodeB (gNB) when operating in environments where power supply is scarce or intermittently available, as per [36], but it is unclear if this can include adjustments to the PSS or SSS signal features that were used in this study, as these are essential for UE devices to detect and associate with the gNB or eNodeB (eNB). Some telecoms operators do pursue a power saving strategy involving the de-activation of an underutilized gNB that is co-located with an eNB when the utilization level of the cell does not require the throughput or other capabilities of the 5G radio, as per Shen et al. [37].

While these behaviors have impacts on the passive exploitation of these signals for either navigation or timing, the data presented in this paper did not suffer from the losses of lock caused by dynamic power reduction, nor a loss of tracking due to source deactivation, so the power saving did not appear to impact the results presented. The base stations could, in theory, reduce the power at which the PSS and SSS are transmitted. Given the information available, it is hard to tell if this is, in actuality, implemented in the towers available during the measurement campaign, but the authors believe that this is of no consequence in either case. Power saving could reduce the SNR and thus make acquisition more difficult or lead to a higher rate in the loss of lock; however, in a scenario where no loss of locks occur, such as in this paper, power saving should not have an appreciable effect on the results presented. 

### 4.2. Further Work

This study was initially started as part of a larger effort looking into cellular SoP-based navigation. The results presented here are directly useful for any application considering use of LTE or 5G signals for time holdover or SoP navigation exploitation. One way in which to interpret the information presented in this paper is that a system tracking the signals with a setup comparable to the one presented and measuring the relative ranges to a tower can achieve a time error rate in the order of 0.03 parts per billion (ppb). This can provide additional constraints on existing navigation platforms, particularly for drones operating at altitude, which will be minimally affected by multipath technology, thereby significantly improving the quality of the navigation solution in degraded environments (such as in situations where the GNSS is jammed or otherwise not available). For example, the results obtained from the ICE tower showed that, over longer time frames, the range measurement can drift significantly with some towers, but a short-term range rate provides high-quality results for all observed towers. This is important information for when range or velocity constraints on navigation platforms are considered.

Another way in which this work can be improved is by trying to relax the requirements on the local oscillator. For practical applications, it would be desirable to have a receiver system that does not depend on a GPS-disciplined Rb oscillator. This work provides valuable insight for further steps where it could be investigated if differential measurements from multiple towers can be used to estimate the local clock error in a manner similar to how GNSS state estimation is performed. 

## 5. Conclusions

The implications of this work to the PNT domain are clear and concise in that we have demonstrated that different network infrastructures can exhibit a wide range of variable performance when used for the purposes of SoP navigation, particularly regarding velocity constraints or time holdover purposes. The question of which infrastructure should be used and even at what time scale the measurements should be taken for optimal performance varies with tower-specific implementations, and this fact should be taken into account when the signals are passively exploited for PNT purposes. A weak inference based on several differential pairs tested is that a measurement update rate of approximately 100 Hz might be selected as a default measurement rate in the absence of other information about the tower or tower pairs when one is using it for SoP navigation (as longer observation intervals are past the knee of the Allan deviation curve). Further testing is needed to determine if this is generally true, as well as to what extent lower cost local oscillators might contribute to in masking this effect.

## Figures and Tables

**Figure 1 sensors-24-00213-f001:**
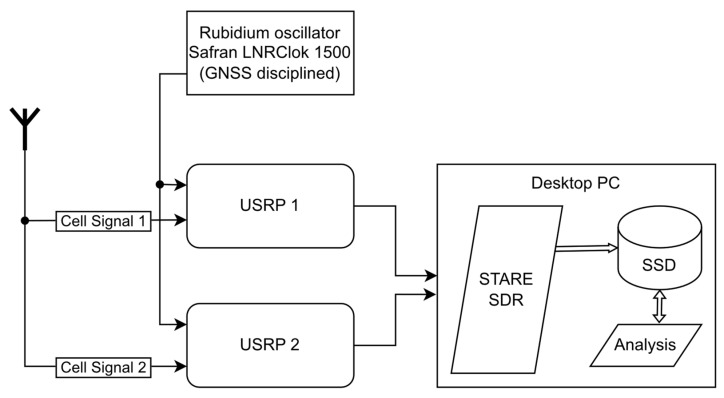
Conceptual overview of the experimental test setup.

**Figure 2 sensors-24-00213-f002:**
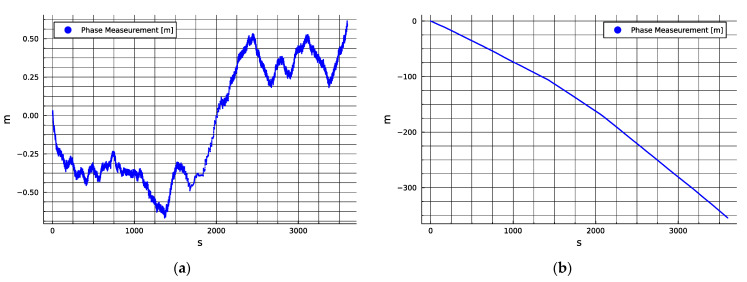
(**a**) This station demonstrates a peak-to-peak one hour deviation of less than 1.25 m, which is equivalent to a time delta of 4.2 nanoseconds; (**b**) this station demonstrates a much higher level of deviation equivalent to more than 1200 ns.

**Figure 3 sensors-24-00213-f003:**
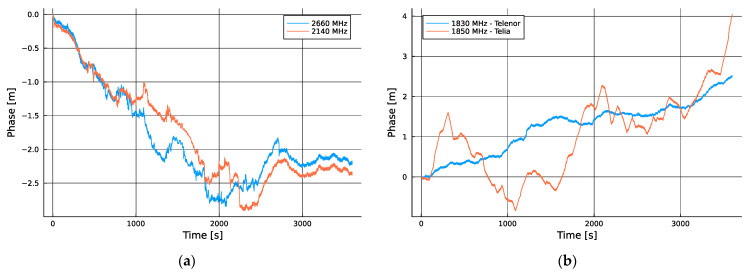
(**a**) Phase measurement evolution of two Telenor cells; (**b**) phase measurement evolution of one Telenor cell and one Telia cell.

**Figure 4 sensors-24-00213-f004:**
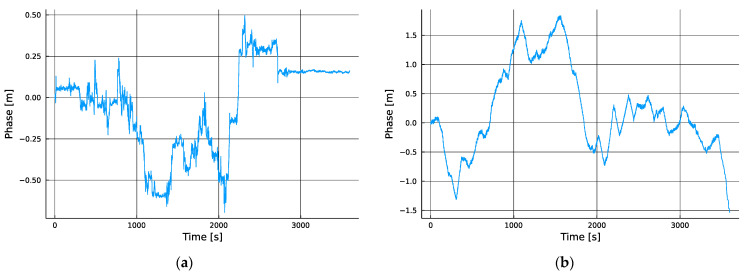
(**a**) The delta between the two Telenor towers was bounded to a 1.25-m peak-to-peak range over the course of one hour, but exhibited step changes; (**b**) the delta between a Telenor and Telia tower reached a 3.5 m peak-to-peak range over the hour.

**Figure 5 sensors-24-00213-f005:**
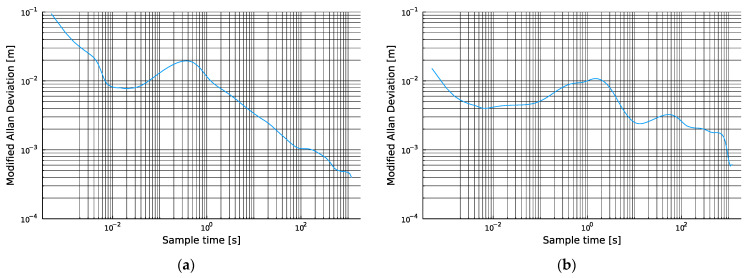
(**a**) Phase measurement instability between two Telenor cells; (**b**) phase measurement instability between one Telenor cell and one Telia cell.

**Table 2 sensors-24-00213-t002:** Common settings for all the recordings discussed in this paper.

Property	Value
PLL Loop Bandwidth	5 Hz
DLL Loop Bandwidth	2 Hz
FFT Size	1024
Subcarrier Spacing	15 kHz
Complex Sample Rate	15.36 MS/s
USRP Frontend Gain	40 dB

## Data Availability

Data available on request due to size considerations.

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
