# Peer review of "Analysis of 5G and LTE Signals for Opportunistic Navigation and Time Holdover"

_sensors, 2023, doi:10.3390/s24010213_

Round 1

Reviewer 1 Report

Comments and Suggestions for Authors

The paper investigates the suitability of cellular signals for PNT. It is well written and interesting overall. The authors are suggested to consider minor issues below 

-- lines117-119: multipath between static transceiver may not be constant due to moving scatterers; authors are suggested to reconsider this assumption which seems to be important in the experiment designs. 

-- phase measurement in Fig. 2 is in unit metre; author are suggested what conversion is used here ?

-- Fig. 2(b) may be caused by phase unwrapping? can authors double check?

Comments on the Quality of English Language

fluent and comfortable

Author Response

Dear reviewer, thank you for your feedback!

Based on the feedback of the other reviewer, we have made extensive modifications to the draft. To address your points: 

-- lines117-119: multipath between static transceiver may not be constant due to moving scatterers; authors are suggested to reconsider this assumption which seems to be important in the experiment designs. 

Thank you for pointing this out, this is indeed something we thought we had mentioned, but apparently didn't include after all. We added clarifications to Sections 2.2 and 3.2 discussing moving multipath reflectors. 

-- phase measurement in Fig. 2 is in unit metre; author are suggested what conversion is used here ?

Internally, the software measures time. This is converted to meters by multiplying with the speed of light. A clarification of that was added to Sec. 3.2

-- Fig. 2(b) may be caused by phase unwrapping? can authors double check?

In the authors opinion, this is not caused by a phase unwarpping artefact, because the associated doppler error residual is commensurate with the accumulated range shown.  

We hope this addresses your points and looking forward to any further feedback. 

Reviewer 2 Report

Comments and Suggestions for Authors

Suggestions and comments:

1) Title – do check how to properly use Capital letters and make necessary corrections.

2) Affiliation – if possible, provide full name of each institution.

3) When using abbreviations/acronyms, always provide their full description, both in the abstract as well as main body of the manuscript, when first mentioned.

4) To start with, the number and scope of cited references is far too short, both in the theoretical as well as research part. Do look for surveys and studies focused on, e.g., navigation/positioning methods, 4G and 5G solutions, transmission of different types of content, particularly multimedia, signal propagation in different environment, not to mention GNSS-related topics, as well as mobile devices, terminals, and user equipment. Discuss current trends as well as various quality aspects, including QoS, QoE, UX, etc.

5) The novelty and contents of your paper should be adequately highlighted in the first section. Do use bullet points, and point out what distinguishes your study from previously published ones. Justify why is it important, how can this be utilized in everyday situations, etc.

6) Provide at least principle technical specs considering the utilized USRP (was it Ettus or National Instruments?), as well as used simulation environment, toolboxes, libraries, open-source or custom-build software, etc. Next, what frequency bands, bandwidth, types of content, frame structure, sampling frequency, bitrate, etc., was set? Did you perform your measurements in a lab or open space? How many cellular networks, GNSS signals/constellations, etc., were analyzed? The Materials and methods section is simply far too short.

7) Next, considering the deployment of cellular network operators on the terrain of the Campus – how big was this terrain? Were there any buildings or other natural or man-made obstacles that could influence the signal propagation? Authors should present a figure describing this outdoor environment.

8) Furthermore, how many operators were available in Norway at that time? How many BTSs were around this terrain? What was the operating frequency, transmission mode, etc.? Several important information are missing. Table 1 includes some partial information, but it should be reconfigures. Start with arranging it according to the operator’s name. Were some BTSs shared by 2 or 3 operators? Do they share antennas, bandwidths? Which one was a MNO, and which was only a virtual MNO? Additional comments seem necessary.

9) How long did the measurement campaign last? At what times of the week, what times of the day? How many sessions did it include? How long did a single session last? You are aware that factors, such as the transmission power, is strictly dependent on the time of the day, related with the number of active simultaneous users/equipment, etc.

10) I get the idea that the Authors have decided to evaluate and test cellular networks in the nearest vicinity of the Campus. You did utilize a USRP device to obtain and analyze those signals, freely available. Was this USRP operating as an oscilloscope, etc., or was it programmed to act as a mobile terminal (e.g., cell phone)? Additional comments seem necessary.

11) The discussion and conclusions section are far too short. Do provide additional comments on your findings, provide feedback for the potential reader, mention about open issues and future study directions, etc. The current form is not convincing at all.

To sum up, I do get the general idea behind this paper, yet its current form is not acceptable. Therefore, Authors are strongly advised to extend and modify their manuscript, and preparing a revised version. This could be a good paper, but it requires major revisions.

Author Response

Dear reviewer, 

thank you very much for your comprehensive and valuable feedback. We have attached a PDF document summarizing all of the additions and changes based on your feedback. 

We hope that this satisfies your concerns and we look forward to future communication. 

Round 2

Reviewer 2 Report

Comments and Suggestions for Authors

Authors have prepared a revised version of their manuscript, with respect to the list of suggestions and comments. Obviously, they have the required knowledge and background, the contents seem reasonable and properly justified. As a telecom engineer, I have to say that I am not fully satisfied with the changes made. I am aware of limitations and restrictions related to providing info concerning the MNO infrastructure and the Campus environment. Yet, brief info could be always provided. The list and scope of cited references is also a bit short and limited.

However, this paper fulfils necessary requirements in order to be accepted and published. Therefore, it may be processed further.

Author Response

Dear reviewer, 

thank you again for your feedback, helping us improving our paper, and acceptance.

Kind regards and happy holidays!